# Neuropsychiatric and Psychological Symptoms in Patients with Lyme Disease: A Study of 252 Patients

**DOI:** 10.3390/healthcare9060733

**Published:** 2021-06-14

**Authors:** Finja Hündersen, Stefanie Forst, Erich Kasten

**Affiliations:** Medical School Hamburg, University of Applied Sciences and Medical University, Am Kaiserkai 1, 20457 Hamburg, Germany; steffi.forst@googlemail.com (S.F.); erikasten@aol.com (E.K.)

**Keywords:** borreliosis, Lyme disease, *Borrelia burgdorferi*

## Abstract

This study examined the relationship between neuropsychiatric and psychological symptoms in patients with Lyme borreliosis. We collected data from an experimental group of 252 Lyme disease patients and a control group of 267 healthy individuals. The quality of life and sleep, attention and memory performance were assessed in both groups. Additionally, we investigated depressive symptoms in patients with Lyme disease to examine whether the duration of the disease had an influence on the severity of symptoms shown. Furthermore, various data on the diagnostics and treatment carried out in the patient group were recorded. On average, patients visited almost eight physicians to obtain a diagnosis, and eight years passed between the tick bite and diagnosis (SD ± 7.8); less than half of the sample (46%) received their diagnosis within the first five years after the development of symptoms. It became clear that Lyme disease is often diagnosed very late. It appears that people suffering from Lyme disease have significantly lower quality of life and sleep and show cognitive impairments when it comes to attention and memory. This study shows that 3.1% of Lyme patients were satisfied with their lives and that 37% scored in the lower third of the quality-of-life scale. It was also shown that Lyme patients tend to have depressive symptoms.

## 1. Introduction

Lyme disease is an infectious disease caused by ticks infected with the bacterium *Borrelia burgdorferi* [1]. These bacteria can be detected in 5 to 35 percent of ticks in Germany, depending on the area [2]. The disease of Lyme borreliosis manifests through diverse and varying symptoms, which make recognition and treatment considerably difficult [3]. In general, Lyme disease is a worldwide growing health problem in countries of the northern hemisphere. In addition to the general symptoms of Lyme disease, which can affect the skin, musculoskeletal system, central and peripheral nervous system, as well as cardiac muscles and eyes [4], psychological and neuropsychiatric symptoms are also present. In addition, Lyme disease poses an economic problem: due to physical limitations or frequent visits to the doctor, affected individuals cannot maintain their usual work performance and working hours, often leading to an adjustment or termination of their job contract [5]. Furthermore, the illness is the cause of significant costs for the healthcare system: for example, high costs are incurred for diagnostic tests and treatments [6]. In addition to somatic complaints, the following psychological impairments can occur as a result of the infection: mood swings; depression; personality disorders; and problems with sleep, concentration, thinking, and memory [7]. It remains unclear whether mental health problems, such as depression, occur comorbidly or develop as a consequence of dealing with an unpredictable disease. According to Fallon et al. (1996), for example, the fact that mental health symptoms decrease after successful antibiotic treatment could indicate that depression is a direct consequence of Lyme disease [8].

It often takes months to years and various visits to the doctor before a proper diagnosis of Lyme disease can be made. Prior to diagnosis, the symptoms are often falsely considered to be psychologically caused, which means that many patients are treated with psychopharmacological treatment or psychotherapy, usually without success. This leads to an advanced stage of the disease, which in turn complicates the therapy of Lyme borreliosis. In addition, there is the high burden of nerve toxins, which severely limits not only the physical but also the mental resilience of those affected. Such patients lack the energy to cope with everyday life, are unable to work for a long time, and often retire early. Furthermore, patients often withdraw from their social environment; their changeable symptoms make them feel insecure and make it difficult for them to plan their daily lives. The situation is further aggravated by the fact that there is a lack of awareness in the patients’ social environment. Due to the multitude of psychological impairments described above, support from a psychotherapist is highly recommended [7].

In the present study, various psychological symptoms were recorded in a group of Lyme patients and compared with a control group. These included quality of life, sleep, attention, and memory performance; and depressive symptoms.

## 2. Materials and Methods

The conducted study included a sample of *n* = 523 subjects. There were *n* = 252 subjects in the experimental group (Lyme disease patients) and *n* = 267 in the control group (non-sufferers). Data were collected using an online questionnaire via the survey software Unipark. At the beginning of the questionnaire, the ethical consent of the subjects was obtained and they were informed about the data processing. To reach the widest possible sample, the link to this survey was shared in several Lyme disease specialized Facebook groups. In addition, a flyer was created and sent out in online newsletters of medical practices as well as Lyme disease information sites. Additionally, the flyer was displayed in the offices of physicians who specialize in the treatment of Lyme disease. The control group was also approached via social networks, such as Facebook. Furthermore, the test person system of the Medical School of Hamburg was used. The data collection period was from February to April 2020 inclusive.

The questionnaire used consisted of different items, which were taken from existing questionnaires as well as self-constructed items. It should be noted that the originally developed version, with several standardized test procedures, had to be shortened considerably, given that preliminary tests showed that the group of Lyme disease patients were clearly overloaded by the sheer length of the survey. In addition to demographic data, varied information on the diagnostic and treatment process was requested from the group of patients with the disease. The QoL-1 was used to assess quality of life [9], and limitations in memory were captured by several items of the Everyday Memory Questionnaire (EMQ) [10]. For sleep quality assessment, some items of the Pittsburgh Sleep Quality Index (PSQI) [11] as well as self-constructed items were presented. To assess subjects’ subjective attentional performance, some items from the Scale for the Assessment of Attention Deficits (SEA) were used [12]. Depressive symptoms were assessed using the simplified version of Beck’s Depression Inventory [13] in the group of sufferers. In addition, the questionnaire contained questions requesting information about the diagnostic and therapeutic process. The items that elicited information about antibiotic therapy were used from an existing survey [14]. For example, participants were asked whether there had been a relapse after antibiotic therapy. Relapse was not defined in more specific terms in this case, as it was assumed that patients understood a relapse of symptoms to be a relapse of Lyme disease. The data were analyzed using the statistical program SPSS.

### 2.1. Sample

The experimental group included 189 women, 62 men, and 1 non-binary person. The average age of the experimental group was 51 years, with an age range of 19 to 84 years. The control group consisted of 211 women and 56 men; the average age of the control group was 27 years, with an age range of 18 to 70 years (see Table 1 and Table 2). The inclusion criterion for the experimental group was the presence of Lyme disease. Specific exclusion criteria were not defined.

Since the two samples differed significantly in terms of age, a correlation was used to determine whether there was an age effect. For this purpose, a correlation of age with each individual variable of the sample was carried out. It was found that age had no significant influence on the variables. In addition, all variables were tested for a normal distribution. Since this could be ruled out, all results were calculated using non-parametric procedures. In principle, a significance level of five percent was used. Since some items from the questionnaires used were reformulated or self-formulated items, a reliability analysis was performed. In addition, a factor analysis was performed for all scales for which improvements were minimal, which meant that excluding the items was not considered.

### 2.2. Sample Characteristics

#### 2.2.1. First Symptoms and Tick Bite

Those subjects who could recall the triggering tick bite were asked how much time elapsed until the diagnosis of Lyme disease. Just under half (46.4%) received a diagnosis of Lyme disease after a period of 0 to 5 years after the tick bite. A minor proportion of subjects (1.8%) waited 34 to 40 years for a proper diagnosis. The overall mean was 8 years, with a standard deviation of 7.7 years, and the median was 6 years (IQR: 2 to 13).

A total of 187 individuals (74%) could recall a tick bite that possibly triggered Lyme disease.

Nearly 45% (45.5%) of all subjects recalled an erythema migrans.

#### 2.2.2. Number of Physicians

On average, patients visited nearly 8 physicians (7.8) to obtain a diagnosis. The standard deviation is 11 physicians, with a median of 6 (3 to 10).

#### 2.2.3. Co-Infections

In addition, co-infections were investigated in the experimental group. The majority of the subjects stated that they had no co-infections or had not been tested for any. A total of 25% of the sample stated that they had (exactly one) co-infection. Two co-infections were found in 15% of the sample. For patients that suffered from co-infections, chlamydia infection was particularly common (34.9%), as was the Epstein–Barr virus (34.9%). In just under a third of the respondents (31%), no diagnosis of co-infections had been made according to the information provided by the persons concerned.

#### 2.2.4. Antibiotic Therapy

The majority of subjects (38.1%) stated that they had paid part of the costs of antibiotic therapy themselves. The difference between those who paid the costs alone (15%) and those who did not (14%) was small. In addition, in about 8% of the cases, the costs amounted to either less than EUR 1000 and up to EUR 2000. In 7.1% of the cases, the costs ranged from EUR 10,000 to 20,000.

In total, 168 patients received antibiotic therapy over a period of more than three weeks. The type of treatment was oral in most cases (47.6%), or a combination of oral and intravenous therapy (48.8%); in rarer cases, an exclusively intravenous therapy (3.6%) was undertaken. Nearly 73% (72.9%) of subjects that were treated with antibiotic therapy for longer than 3 weeks showed an improvement in symptoms. Overall, however, 93% (92.9%) reported a relapse of symptoms after the completion of antibiotic therapy.

#### 2.2.5. Diagnostic Tool

Multiple selection was used to determine under which procedure each patient received their diagnosis. Among others (several methods were given for selection), the antibody test enzyme-linked immuno-assay (ELISA), Western blot, lymphocyte transformation test (LTT), tick-plex, as well as a diagnosis by sighting of the skin disease acrodermatitis chronica atrophicans (ACA) and an examination of the cerebrospinal fluid were available for selection.

In general, a two-tiered diagnostic procedure is recommended for the diagnosis of Lyme borreliosis. It consists of a screening and a confirmatory test. For the screening test, a sensitive ELISA is recommended, which differentially detects antibodies. In cases of a positive or indeterminate result, a confirmatory test—an immunoblot—should be performed [15].

ELISA can detect immunoglobulin M (IgM) and immunoglobulin G (IgG) antibodies in the blood, mapping the immune system response [3]. IgM is produced during an acute initial infection, leading to an increase in IgM titer in the blood. Therefore, a recent infection can be detected by an increased IgM level. As infection progresses, IgG antibodies increase. Thus, a progressive infection can be detected by an elevated IgG titer [16].

Western blot is considered a screening method: either full antigens from pure cultures of *Borrelia* strains or genetically produced, clearly defined *Borrelia* are used. These are applied to a blot membrane to react with the IgM and IgG antibodies of the patient’s blood. After this particular reaction, the proteins are separated, resulting in an electric field, the *borrelia*-specific bands detectable for the patient. Western blot is used to confirm the first step, i.e., the positive test for antibodies. The antibody pattern found can be further analyzed, allowing conclusions to be reached regarding the age of infection. In this way, early and late bands can be identified, and information can be provided about how long the infection has been present.

The lymphocyte transformation test (LTT) is another immunological examination method. The immune response of T-lymphocytes to a contact with *Borrelia* antigens is measured. The main advantage of the LTT is that a result can be determined as early as 10 days after infection, although the test result takes a week. The test value remains positive as long as T memory lymphocytes are still activated by the *borrelia* in the body of the affected person, which means that the LTT also proves valuable in later stages of the disease [3]. Although the LTT is currently not recommended by any guideline due to its low specificity [17], we nevertheless asked about it, as it is frequently used in German clinical practice.

The tick-plex assay is a procedure based on an ELISA. However, what is different is that a new antigen for more persistent forms of *Borrelia* is included. IgM and IgG antibodies of several bacterial and viral pathogens can be determined simultaneously [18]. Nevertheless, the diagnostic accuracy of ELISAs for Lyme disease varies widely across Europe. An average sensitivity of approximately 80% and a specificity of approximately 95% is assumed [19].

Acrodermatitis chronica atrophicans (ACA), a typical skin disease that can occur as a result of a *borrelia* infection, is characterized by the transformation of atrophy of the skin and subcutaneous tissue on the hands and feet. It usually occurs on one side and only rarely on both sides [3].

If neuroborreliosis is suspected, examination of the cerebrospinal fluid is recommended. Therefore, the *borrelia*-specific CSF serum antibody index can be calculated [20].

In the study conducted, it was found that the majority of patients were diagnosed by ELISA (53.2%), Western blot (43.7%), and LTT (41.7%). A minority of patients were diagnosed by tick-plex (0.8%), ACA (4.4%), and CSF examination (9.1%).

## 3. Results

### 3.1. Quality of Life

The main results showed that the quality of life was worse in the Lyme disease group compared to the control group. The Mann–Whitney U test was significant, and a large effect could be calculated using Cohen’s effect size (U = 9840, Z = −14, *p* < 0.001, r = −0.6). The life satisfaction scale ranged from 1 to 11, with 1 being labeled as “not at all satisfied” and 11 as “completely satisfied”. The mean score for the experimental group was 4.9 with a standard deviation of 2.7, and for the control group, it was 8.4 with a standard deviation of 1.6.

### 3.2. Sleep Quality

Sleep was recorded and evaluated using several items, and an overall impaired sleep quality was found in the group of sufferers (see Table 3).

### 3.3. Attention

Attention was assessed using a selection of items from the SEA. The items were answered using a Likert scale including 0 (never), 1 (rarely), 2 (sometimes), 3 (most of the time), and 4 (always). Therefore, the mean score in the experimental group was 3.2 with a standard deviation of 1.1, and in the control group, it was 1.2 with a standard deviation of 0.7. In addition, statistically significant differences compared to the control group as well as a strong effect (U = 9031.5, Z = −14.6, *p* < 0.001, r = −0.6) were found.

### 3.4. Depressive Symptoms

Depressive symptoms were assessed using the simplified version of the BDI (BDI-V: Schmitt, Weinheim: Germany). Since the test measures the severity of existing depression, only the experimental group was questioned and compared with the norm. The mean value was 45.1 with a standard deviation of 20.5. Clinically relevant depression can be assumed at a value of 35 or higher. In the questionnaire, items were presented and could be confirmed or rejected based on five different response options. The answer categories ranged from 0 (never) to 5 (almost always). In total, 68% of the experimental group showed scores above 35 and thus exhibited depressive symptoms.

### 3.5. Memory Performance

Memory performance was determined using some items of the EMQ. A statistically significant difference was found between the two groups, with a mean score of 2.8 (SD = 0.996) in the experimental group and 1.8 (SD = 0.6) in the control group on a scale of 1 (the statement was rarely true) to 4 (the statement was true several times a day). Moreover, a strong effect was recorded (U = 0.001, Z = −20.4, *p* < 0.001, r = −0.9).

### 3.6. Career Situation

Subjects were asked about their work/school situation, and almost 10 percent (9.9%) of the respondents reported being unable to work due to Lyme disease. A disability pension is received by 26 percent of the respondents. In contrast, in November 2020, around 44.5 million people were registered as being in employment in Germany. According to the German pension insurance fund, around 1.8 million people received a pension for reduced earning capacity in July 2020, which is under four percent.

## 4. Discussion

The results of the conducted study show that Lyme disease can cause diverse psychological and neuropsychiatric symptoms. These include limitations in quality of life, sleep, attention, and memory, as well as depressive symptoms.

In addition, it was found that Lyme patients have to wait an average of 8 years and see an average of eight physicians before a final diagnosis is made. Less than half of the subjects (46%), received their diagnosis within the first 5 years after the development of their symptoms. This highlights the amount of time and money that must be spent to obtain a diagnosis. The most common methods for diagnosing Lyme disease in the present study are ELISA, Western, blot and LTT. One co-infection was found in 25.1% of the subjects, and two co-infections were found in 15%. In 30.5% of the cases, no diagnosis was made in this regard. Erythema migrans was remembered by only 45.5% of the present sample. Almost 73% of the subjects reported an improvement of symptoms after the implementation of a three-week antibiotic therapy; however, 93% suffered a relapse afterwards.

Overall, the study shows an up-to-date insight into the clinical picture, particularly in relation to psychological and neuropsychiatric symptoms, the diagnostic process, as well as the treatment with antibiotics.

Nevertheless, it must be noted here that the study involved self-reported data. For example, it must be taken into account that people suffering from memory loss may not be able to accurately remember appointments with doctors (date and/or content). Self-assessment of memory and attention is widely used in research; however, there are many studies showing that the results of such self-assessments are less specific than neurocognitive tests. Therefore, it is recommended to use neurocognitive tests in future research to assess memory and attention performance. An option for validation could be a comparison of the self-reported statements with the patient’s file, which could either verify the accuracy of the patients’ statements or reveal a poorer memory performance of the patients. Another issue is the generalizability of a survey-based study. The question arises as to how far the data collected can be transferred to the population. Nevertheless, it should be noted at this point that the sample contained a total of over 500 subjects. However, it is not clear whether the included participants underwent a two-tier diagnostic, as recommended by several guidelines. Another limitation is the inclusion of the LTT. As already mentioned, the LTT is not recommended for diagnostic purposes by the majority of guidelines due to its low specificity. In the present study, it was included because it is frequently used in German clinical practice. This should definitely be taken into account when considering the data.

The results presented are in line with existing publications. This can be seen, for example, in the quality of life [5,21,22,23] and in the varied information collected related to the diagnostic process [3,6,24]. In summary, the present research was able to establish significant associations between the investigated symptoms and chronic Lyme disease. Treatment approaches should be developed that support the patient psychologically in addition to the medicinal symptom treatment of Lyme disease.

## Figures and Tables

**Table 1 healthcare-09-00733-t001:** Age: experimental group.

Age	Count	Percentage
Under 30	17	6.7
31–50	100	39.7
51–70	119	47.3
>70	16	6.3

**Table 2 healthcare-09-00733-t002:** Age: control group.

Age	Count	Percentage
Under 30	232	86.9
31–50	18	6.7
51–70	17	6.4

**Table 3 healthcare-09-00733-t003:** Sleep quality in the experimental and control groups.

Item	Mean ± Standard Deviation	*p*, U-Value, z-Value	R (Effect Size Cohen’s d)
Tired in the afternoon (scale range from 0 to 10)	Experimental group: 7.7 (±2.7) Control group: 5.1 (±2.4)	*p* < 0.001 U = 15,736 z = −10.5	−0.5
In need of taking a nap (scale range from 0 to 10)	Experimental group: 5.8 (±3.7) Control group: 3.4 (±3)	*p* < 0.001 U = 20,975 z = −7.6	−0.3
Tired after getting up (scale range from 0 to 10)	Experimental group: 7 (±3.2) Control group: 3.9 (±2.7)	*p* < 0.001 U = 15,728 z = −10.5	0.5
Sleeping for several hours a day (scale range from 0 to 10)	Experimental group: 3.6 (±3) Control group: 2.1 (±1.9)	*p* < 0.001 U = 22,849 z = −6.7	−0.3
Sleep-wake rhythm confused (scale range from 0 to 10)	Experimental group: 4.8 (±3.5) Control group: 3.2 (±2.8)	*p* < 0.001 U = 24,987 z = −5.2	0.2
Subjective evaluation of sleep quality (Scale from 1 (very bad) to 4 (very good))	Experimental group: 2.8 (±0.8) Control group: 2.2 (±0.7)	*p* < 0.001 U = 19,142 z = −9.2	0.4

## Data Availability

Not applicable.

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
