# Peer review of "Neuropsychiatric and Psychological Symptoms in Patients with Lyme Disease: A Study of 252 Patients"

_healthcare, 2021, doi:10.3390/healthcare9060733_

Round 1

Reviewer 1 Report

This is a very interesting study which will be useful in prompting improvements in diagnosis and treatment.  

The following points should be addressed:

Please describe the recruitment process, including information about  who the participants in the control group are and how they were invited to participate?

There is no mention of ethical approval for this study.  This is essential.

Data in this study is self-reported.  This should be clearly specified and acknowledged as a potential weakness in the study.  For example, individuals who have symptoms of memory loss may not be able to accurately recall appointments with clinicians (date or content).  If this data could be matched with patient records, this would serve to validate self-reported statements or expose incidents of poor recall by participants.  

Phrases such as "it turned out", "carrying out sleeping" should be improved.  Improvements also needed on lines 71, 78-80, 215-218, 233.

The age of the youngest participant is 13.  I think that no participants under the age of 18 should be included as (a) it is unclear if ethical approval was obtained for data collection from children and (b) self-reported symptoms and recall of appointments with clinicians is problematic.  At the very least, if children are to be included, their data should be treated / reported separately.

There is a lack of reference material included in the section 39-54.  The statements which have been made need to be backed up.

Is there a database of results available?  This is good practice.  More data could also be included as appendices.

Author Response

First of all, thank you for your detailed review and helpful comments. 

I added the recruitment process, and defined the control group and how they were recruited.

Ethical approval is added at the end of the manuscript as the informed consent. 

Thank you very much for the helpful advice that the data are self-reported and therefore also have weaknesses. I have taken this on board, of course. 

I have adjusted the formulations mentioned. 

I have also adjusted the groups and removed the subjects under 18 years and recalculated the results. 

Also I added a bit more reference material at the suggested part of the text. 

No unfortunately theres no database.

Thank you very much for your help. 

Reviewer 2 Report

This paper is a case-control study evaluating the cognitive and affective burden in patients with a positive laboratory test for B.burgdorferi exposure or, in a small minority, a suggestive clinical condition associated to late Lyme disease (acrodermatitis chronica atrophicans). Writing needs deep proofreading for typos and english.

My preminent concern is the complete lack of definition of the population being investigated. Cases are defined as subjects with a positive laboratory test, but not why the patient had been tested at all.

This is utterly important: current guidelines (10.1093/cid/ciaa1215) stress the importance of testing only patients with specific clinical presentation as erythema migrans, neuroborreliosis strictly defined as "meningitis, painful radiculoneuritis, mononeuropathy multiplex including confluent mononeuropathy multiplex, acute cranial neuropathies , or in patients with evidence of spinal cord -or rarely brain- inflammation", Lyme carditis, Lyme arthritis, borrelial lymphocytoma.

Moreover, current guidelines almost unanimously recommend a two-tier diagnostic approach, using a serological test first, confirmed by immunoblot or PCR (10.1016/j.medmal.2018.11.011). Lymphocyte Transformation Test is not currently supported by any but one guideline due to low specificity (10.1016/j.cmi.2019.06.033).

If available, provide the specific antibiotics regimen and the criteria you used to define a "relapse". 93% failure rate is extremely high and much different from previous studies (10.1128/aac.39.3.661 ; 10.7326/0003-4819-117-4-273)

Authors should explain their method of enrollment along with explicit inclusion/exclusion criteria, and describe the relevant clinical characteristics in their sample.

Minor considerations

  • data presented under headings Sample, First symptom and tick bite, Number of physicians, Co-infections, Antibiotic therapy, Diagnostic tool are better suited to the Results sections under a "Sample characteristics" heading. Similarly, you should give the definition of the attention, depression and memory test in Methods and not in Results
  • As a non-binding opinion, I would shorten the chapter on diagnostic tool testing as it seems unnecessary verbous considering they are diffusely used tests. Also, I would refrain from citing only a particular commercial ELISA testing kit
  • please provide values as median (IQR), as variables did not follow a normal distribution
  • round up numbers consistently, either by keeping only cyphers before the zero or one digit after the zero
  • In adult population, EBV has a seroprevalence of 80% and Chlamydia 4-5%. Given that neither microorganisms are tick-borne, the authors should discuss these findings and their significance for their study group
  • Antibiotic therapy: needs rephrasing as it is difficult to understand what the percentages are referring to, more so as the numbers do not adds up to 100%
  • Sleep quality: "overall impaired sleep quality was found in the group of sufferers", however in Table 3: "Subjective evaluation of sleep quality - Scale from 1 (very bad) to 4 (very good)" "Experimental group 2.77 ; Control group 2.15"
  • References: given that there are is plenty of english literature available on borreliosis and the journal is aimed to an international audience, I would try to find references from the english literature (10/20 references in German)
  • Title typo: symptoms
  • line 9 and line 25: adjective is lower case and normal: Lyme borreliosis. Species is upper case and italics: Borrelia burgdorferi
  • line 84: please change "1 miscellaneous person" to "non-binary"

Author Response

First of all, thank you for your detailed review and helpful comments. 

I have added more information to clearly define the investigated population. I am not sure what you mean by the positive laboratory test. We did not tested the patients, it’s all self-reported data. Also I added some points to clarify that. 

Thank you for pointing out to improve the part regarding the diagnostic of Lyme and use a better source for the ELISA. 

I improved the antibiotic part.

Also I improved the recruiting process of the population, including the inclusion/exclusion. 

Furthermore your heading „Minor Considerations“ have been very helpful.

Again thank you so much for your review. 

Reviewer 3 Report

I would discuss the difficulties generalizing a survey-based study. 

I would also discuss the difficulty determining whether the depression is a neuropsychiatric manifestation of Lyme disease, is related to their poor quality of life, or an a co-morbidity.

Author Response

First of all, thank you for your detailed review and helpful comments. 

I added a part where I discussed the difficulties generalizing a survey-based study. 

I also added a paragraph to discuss what the depression in Lyme disease develops from. 

Again thank you so much for your review. 
